# Towards Novel Potential Molecular Targets for Antidepressant and Antipsychotic Pharmacotherapies

**DOI:** 10.3390/ijms24119482

**Published:** 2023-05-30

**Authors:** Yuriy M. Kositsyn, Murilo S. de Abreu, Tatiana O. Kolesnikova, Alexey A. Lagunin, Vladimir V. Poroikov, Hasmik S. Harutyunyan, Konstantin B. Yenkoyan, Allan V. Kalueff

**Affiliations:** 1Institute of Experimental Medicine, Almazov National Medical Research Centre, Ministry of Healthcare of Russian Federation, St. Petersburg 197341, Russia; yuriikositsyn@gmail.com (Y.M.K.); philimontani@yandex.ru (T.O.K.); 2Neurobiology Program, Sirius University of Science and Technology, Sirius Federal Territory 354340, Russia; 3Institute of Translational Biomedicine, St. Petersburg State University, St. Petersburg 199034, Russia; 4Laboratory of Preclinical Bioscreening, Granov Russian Research Center of Radiology and Surgical Technologies, Ministry of Healthcare of Russian Federation, Pesochny 197758, Russia; 5Neuroscience Group, Moscow Institute of Physics and Technology, Moscow 115184, Russia; abreu_murilo@hotmail.com; 6Vivarium, Ural Federal University, Yekaterinburg 620049, Russia; 7Department of Bioinformatics, Institute of Biomedical Chemistry, Moscow 119121, Russia; alexey.lagunin@ibmc.msk.ru (A.A.L.); vvp1951@yandex.ru (V.V.P.); 8Department of Bioinformatics, Pirogov Russian National Research Medical University, Moscow 117997, Russia; 9Neuroscience Laboratory, COBRAIN Center, Yerevan State Medical University Named after M. Heratsi, Yerevan 0025, Armenia; hasmikharutyunyan28@gmail.com; 10Department of Biochemistry, Yerevan State Medical University Named after M. Heratsi, Yerevan 0025, Armenia

**Keywords:** depression, psychosis, schizophrenia, pathogenesis, novel molecular targets

## Abstract

Depression and schizophrenia are two highly prevalent and severely debilitating neuropsychiatric disorders. Both conventional antidepressant and antipsychotic pharmacotherapies are often inefficient clinically, causing multiple side effects and serious patient compliance problems. Collectively, this calls for the development of novel drug targets for treating depressed and schizophrenic patients. Here, we discuss recent translational advances, research tools and approaches, aiming to facilitate innovative drug discovery in this field. Providing a comprehensive overview of current antidepressants and antipsychotic drugs, we also outline potential novel molecular targets for treating depression and schizophrenia. We also critically evaluate multiple translational challenges and summarize various open questions, in order to foster further integrative cross-discipline research into antidepressant and antipsychotic drug development.

## 1. Introduction

Neuropsychiatric disorders, especially schizophrenia and depression, are a major cause of human disability and a common risk factor of mortality [1]. Conventional antidepressant and antipsychotic pharmacotherapies are widely used to treat these two highly prevalent and severely debilitating disorders [2]. However, despite the growing drug intake and availability globally, such pharmacotherapies are often inefficient clinically, causing multiple effects and serious patient compliance problems. With the rise of clinical prevalence worldwide, depression and schizophrenia, especially their treatment-resistant forms [3,4], are becoming an urgent unmet biomedical problem, necessitating novel drug targets and broader, translationally-based pharmacotherapy.

Depression, a highly prevalent mental illness that affects ~5% of the global population, is characterized by low mood, anhedonia, fatigue, attention deficits, suicidal thoughts, motor retardation and neuroendocrine deficits [5,6]. Caused by both genetic and environmental factors [7,8], it often represents a recurrent pathology [9,10,11] with overt monoaminergic, glutamatergic and gamma aminobutyric acid (GABA)-ergic deficits (Figure 1) [2], and multiple genetic risk factors, such as polymorphisms in the dopamine transporter (DAT) [12] and serotonin transporter (SERT) genes [13].

Schizophrenia (psychosis) is a severe psychiatric disorder that affects ~1% of the global population [14]. It typically presents as ‘positive’ (delirium and hallucinations), ‘negative’ (anhedonia, abulia and alogia), cognitive (impaired learning and planning skills) and motor (e.g., dyskinesia, catatonia and hypokinesia) symptoms [15,16]. The pathogenesis of schizophrenia involves multiple neurochemical deficits, especially within the glutamate-, GABA- and monoaminergic signaling systems [2,17,18] (Figure 1). Patients with schizophrenia often have increased levels of dopamine [19] with reduced glutamatergic N-methyl-D-aspartate (NMDA) receptor and (albeit not always) GABA-ergic activity [20]. Risks of psychosis correlate with higher striatal dopamine D2 receptor occupancy [21], further linking dopamine dysregulation and psychosis [22]. While glutamatergic deficits may provoke negative and cognitive symptoms of schizophrenia [23], the disorder is likely linked to disrupted ontogenesis of the glutamatergic and GABAergic neurons [24], and aberrant dorsolateral prefrontal cortex glutamatergic circuitry [25].

## 2. Pharmacotherapy of Depression

The most commonly prescribed conventional antidepressants include selective serotonin reuptake inhibitors (SSRIs), tricyclic antidepressants and monoamine oxidase (MAO) inhibitors [26,27,28] (Figure 2). They act via several different mechanisms, modulating the uptake, reuptake, synthesis and/or metabolism of neurotransmitters [26], as well as the activity of neuronal receptors and their expression (e.g., stimulating postsynaptic serotonin 5-HT1A, postsynaptic 5-HT1B, 5-HT2B and 5-HT4 receptors, or inhibiting presynaptic 5-HT1A, 5-HT1B, 5-HT2A, 5-HT3 and 5-HT7 receptors [29]); also see [2] for a recent review.

Glutamate is the main excitatory neurotransmitter in the brain. Glutamatergic neurons, distributed widely throughout the brain, express ionotropic N-methyl-D-aspartate (NMDA), α-amino-3-hydroxy-5-methyl-4-isoxazolepropionic acid (AMPA) and kainate receptors, and metabotropic G-protein coupled (mGlu) receptors [2]. In the pathogenesis of schizophrenia, NMDA receptors are often downregulated, causing improper glutamate signaling. To reverse impaired functioning of the NMDA receptors and increase the level of glutamate in the synaptic cleft, these neurons likely initiate compensatory events. For instance, while excitatory amino acid transporters (responsible for the reuptake of the glutamate from the synaptic cleft) are downregulated in schizophrenic patients, they also show upregulated glutaminase that converts glutamine to glutamate, in the thalamus and prefrontal cortex [30]. While NMDA antagonists exert antidepressant effects, the glutamatergic, GABAergic and dopaminergic neuronal connectivity overlap [31], hence supporting the clinical link between schizophrenia and depression (Figure 1).

Based on their structure, profile and specificity of ligand binding, metabotropic glutamate receptors are classified into three main groups. Group 1 encompasses mGluR1 and mGluR5, representing G_q_-associated receptors that activate protein kinase C. Group 1 receptor antagonists prevent glutamate from release to the synaptic cleft, thus indirectly reducing its corticolimbic levels, particularly in the amygdala [32]. Thus, the group 1 antagonists exert their antidepressant effects, similar to those of NMDA antagonists. Moreover, mGluR5 antagonists are widely used in animal models of acute and chronic stress. Group 2 includes mGluR2 and mGluR3 G_i_-coupled receptors. Their activation prevents glutamate from the release to the synaptic cleft, and agonists promote depressive episodes [33], likely due to action at the projections to the dorsal raphe nucleus serotoninergic neurons. In contrast, antagonists of mGlu2 and mGlu3 receptors show antidepressant effects. Finally, group 3 includes mGluR4, mGluR6, mGluR7 and mGluR8 G_i_-coupled receptors that prevent glutamate release to the synaptic cleft, and whose agonists demonstrate antidepressant effects in animal models [34]. 

However, no current antidepressants directly target the glutamatergic system except lamotrigine, a phenyltriazine that inhibits glutamate release [35]. Thus, the glutamatergic system can represent a potentially promising novel target for the development of antidepressant agents. For instance, since reduced signaling of glutamatergic neurons may serve as a defensive mechanism to mitigate glutamate toxicity, novel pharmaceuticals that lower glutamate transmission may stabilize plastic changes in the nervous system [36]. Reflecting an important CNS role of glutamate, antidepressants often lower plasma levels of glutamate (that are commonly elevated in depressed patients) [37,38,39,40,41].

Paralleling clinical data [42], animal models of depression also present glutamatergic deficits [31,43] corrected by some antidepressant treatments [44,45]. Disrupted glutamatergic signaling [46,47] is further accompanied by aberrant brain-derived neurotrophic factor (BDNF) and transcription factor cyclic AMP response-binding protein (CREB) signaling [48], with excitatory neurotransmission at ionotropic (AMPA, NMDA) glutamate receptors [49,50]. The glutamatergic system also plays a role in neuroplasticity and neurogenesis via (AMPA)/kainate (KA) receptors and mGluR5, critical for neuronal survival [51,52]. As mounting evidence links depression to aberrant glutamate receptor functioning, glutamatergic drugs (e.g., ketamine and other NMDA receptor antagonists) may be promising as potential multi-target antidepressants [53].

Moreover, NMDA receptor antagonists show consistent antidepressant effects in rodent models [54]. For example, ketamine reduces depression-like states in both animal [55] and clinical studies [56,57,58] while also lowering neuroinflammation, microglia activation and cytokine release in the hippocampus in rodent stress models relevant to depression [59]. Likewise, ketamine lowers lipopolysaccharide (LPS)-induced proinflammatory cytokines interleukin (IL) IL-1β and tumor necrosis factor (TNF)-α in microglia [60]. While anti-inflammatory effects of ketamine are reduced by a colony stimulating factor 1 receptor (CSF1R) antagonist PLX3397, its antidepressant action is modulated by transforming growth factor TGF-β1-dependent mechanisms [61]. Ketamine can also regulate inflammation via toll-like receptors and inhibition of extracellular signal-regulated kinases ERK1/2 [62,63], thus likely modulating affective pathogenesis via neuroimmune mechanisms and circuits (Figure 1). Another mechanism of antidepressant effects of ketamine is the modulation of receptor-mediated effects, since ketamine administration increases signal transducer and activator of transcription 3 (STAT3) levels [64] and the expression of BDNF, synapsin I (SYN1) and postsynaptic density protein 95 (PSD95). Clinical data show that ketamine increases plasma BDNF levels [65] and can also exert antidepressant effects through the mammalian target of the rapamycin (mTOR) signaling system [66], hence impacting neuroplasticity, neuronal survival and synaptogenesis (but see [55]).

GABA is a key inhibitory neurotransmitter [67] acting via GABA-A, GABA-B and GABA-C (GABA a-rho) receptors [68]. GABA-A receptors are ligand-gated ion channels regulating the influx of Cl^−^ ions into neurons. They are an incredibly heterogeneous class of pentameric receptors assembled from multiple subunits (6α, 3β, 3γ, 1δ, 1ε, 1θ, 3ρ) [69]. The hippocampus and cortex receive GABAergic inhibitory inputs that are significantly altered in schizophrenia and depression. Reduced signaling of α5 subunits of GABA-A receptors causes hippocampal hyperexcitation due to insufficient inhibition of glutamatergic neurons and disinhibition of glutamatergic pyramidal neurons, causing loss of synchronous cortical activity and impairments in subcortical dopamine production. Activation of these receptors, in turn, exerts a positive effect on dopaminergic signaling and behavioral aspects in schizophrenic patients [70]. Interestingly, altered expression of various GABA-related genes is observed both in schizophrenia and depression. The former shows predominantly under-expression of GABA-related genes that significantly vary (i.e., increase or decrease) with age [71]. In contrast, the latter is mainly associated with overexpression of GABA-related genes [72], likely with deficient BDNF signaling [73]. Taken together, these findings suggest GABA-A receptors as a promising target for complex multimodal antidepressant therapy.

Mounting evidence suggests inflammation, especially neuroinflammation, as a common risk factor for developing depression. Indeed, depressed patients display higher levels of proinflammatory cytokines [74,75,76], especially TNF-α and IL-6 [77,78]. Neuroinflammation evokes depression-like behavior in rodent models, which is reduced by antidepressants [79,80]. For example, mice after chronic stress develop infiltration of microglia, and increased indoleamine-2,3-dioxygenase (a member of the kynurenine pathway) in the raphe, and TNF-α in the prefrontal cortex [81]. In line with this, the monoclonal antibody infliximab, a TNF-α functional antagonist, lowers symptoms of depression in patients with signs of inflammation, but is ineffective in patients with resistant depression [82].

Moreover, antidepressants can alter the expression of various cytokine genes (e.g., IL-4, IL-6 and interferon gamma (IFN-γ) genes) [83,84,85], while some drugs (e.g., imipramine) downregulate microglia (typically activated in rodent hippocampus after stress) [86]. In rodent models of depression, these drugs may also reduce inflammation [87] and proinflammatory cytokines IFN-γ, IL-6 and TNF-α [88]. Since anti-inflammatory effects of SSRIs can play a crucial role in therapy [89], such multimodal effects of antidepressants in depression merit further scrutiny. However, other antidepressants may exert proinflammatory effects as well. For instance, an SSRI, citalopram, induces TNF-α in brain (corrected by a non-steroidal anti-inflammatory drug ibuprofen) [90], whereas a MAO inhibitor phenelzine triggers neuroinflammation through recruitment of NF-kB [91]. Thus, a more comprehensive and nuanced analysis of both anti- and pro-inflammatory effects of antidepressant drugs is warranted.

Pro-inflammatory cytokines can affect a wide range of neurotransmitter systems (neuropeptides, monoamines, GABA and glutamate) and neuroplasticity processes [92]. Neuroplasticity is a key factor in both affective and psychotic pathogenesis (Figure 1), and potent neurotrophins like BDNF have thereby been probed for their putative therapeutic properties [93,94,95]. The importance of neuroplasticity and BDNF is particularly critical in depression treatment [96,97,98]. For instance, stress may downregulate BDNF in the hippocampus [99], whereas BDNF levels are decreased by pro-inflammatory cytokines [100,101,102]. Glial-derived neurotrophic factor (GDNF) is another key regulator of neurogenesis, whose levels decline in depressed patients [103], but are corrected by antidepressants [104]. The neuropeptide substance P is an agonist for neurokinin-1 (NK-1) receptors, widely expressed in brain regions affected by neuroinflammation. Notably, an NK-1 antagonist orvepitant improves depressive symptoms in clinical trials [105].

Overall, depression is commonly accompanied by brain tissue damage, whereas antidepressant treatment tends to improve neuroplasticity (Figure 1). Moreover, depressed patients often suffer from insomnia, likely representing a comorbid state. Antidepressant effects are shown for melatonin, and the melatonin receptor inhibitor agomelatine is the only antidepressant that corrects the melatoninergic system, also acting as a serotonin 5-HT2C antagonist [106]. Melatonin agonists generally decrease pro-inflammatory processes and promote neurotransmission. Furthermore, opioids are also related to the melatoninergic system, showing striking parallels with the fact that, in animals, the opioid system modulators affect depressive symptomatology, as delta opioid receptor (DOR) agonists [30,31,32,33,34,35,36,37,38,39,40,41,42,43,44,45,46,47,48]) and kappa opioid receptor (KOR) antagonists [54,55,56] exert antidepressant-like effects. Therefore, the link between the melatoninergic system, opioids, neuroinflammation and stress becomes more evident, especially since inflammatory processes can be a core neuropathogenetic factor here, and high concentrations of proinflammatory cytokines may thus diminish concentrations of monoamines and neurotrophins.

Finally, serotonergic psychedelic drugs, currently strictly regulated as hallucinogens in most countries, not only show potential in treatment of psychiatric conditions (e.g., psilocybin in depression [107], also see [108,109]), but also exert immune-modulating effects in vivo as well. Some psychedelic drugs (e.g., psilocybin) have been used to manage treatment-resistant depression. For instance, psilocybin at a single dose reduces depression scores more than a much lower dose given chronically for three weeks [110]. Pramipexole (and, possibly other dopamine agonists) may be useful in treating depression as well, since nearly 80% of treatment-resistant patients show a clinical response to this agent [111]. Similarly, a nutritional adjunctive L-methylfolate (the biologically active form of folic acid, vitamin B9) has also been used [112], increasing clinical responses when co-applied with SSRIs in treatment-resistant depressed patients [113].

## 3. Approaches to Antipsychotic Therapy

As our understanding of schizophrenia and its molecular biomarkers is rapidly growing [114], dopaminergic deficits are strongly implicated in psychotic pathogenesis, especially in its motor, motivation and volition aspects (Figure 1). In general, schizophrenia is presently treated with neuroleptics and benzodiazepines [115,116,117,118,119] (inhibiting dopamine receptors and locomotion), without involving non-dopaminergic drugs as a primary therapy (Figure 2). From the early beginning, dopamine D2 receptors have been targeted in schizophrenia [120], showing higher density in post-mortem brain samples [121] and increased occupancy in patients with higher risks of psychosis [21,122,123]. Additionally, the D2 receptor-adenosine A2A receptor heterodimers seen in basal ganglia, represent a potential target for novel treatment of schizophrenia [121]. Interestingly, cognitive impairments in schizophrenia are associated with hypofunction of the prefrontal cortex, and transgenic mice overexpressing D2 receptors in the striatum show poorer motivation and cognition (e.g., impaired conditioned associative learning) [124], whereas such aberrant phenotypes are rescued by the D2 receptor gene downregulation [125].

As D2 receptors act via both canonical (G-protein-) and non-canonical (beta-arrestin2 βarr2-dependent) pathways, blocking the β-arrestin signaling may evoke antipsychotic effects [126]. The D2/β-arrestin-biased ligands (e.g., UNC9994) are effective in preclinical studies, having an antagonistic influence on D2-βarr2 in prefrontal cortex GABAergic fast-spiking interneurons, yet antagonizing D2-βarr2 in striatal D2 medium spiny neurons, with a dual action likely to prevent hyperdopaminergia [127]. Such ‘dual’ activity is not limited to D2-, but can involve other (e.g., D3 and A2A) receptors as well. Accordingly, additional mechanisms need to be considered for CNS drug development, as they may affect receptors indirectly (e.g., via endocytosis, due to the fact that D2 agonism can induce endocytosis and mediate ligand-based signaling) [128]. Thus, using the β-arrestin-based antagonism with G protein-dependent signaling may hypothetically help reduce positive psychotic symptoms and/or mitigate antipsychotic drugs’ side effects [129].

While NMDA receptors and aberrant glutamate neurotransmission are strongly implicated in schizophrenia [130], some of its deficits may be caused by epigenetic modifications as well. For instance, *RELN* and *GAD1* genes, as well as *NR3B* promotors, are epigenetically modified in schizophrenia [131,132,133], whereas the gene responsible for epigenetic genome modifications (*DNMT1*) is over-expressed in brains of schizophrenic patients [134]. Furthermore, NMDA receptors are downregulated in depressed patients [135], whose positive, cognitive and negative symptoms of schizophrenia are mimicked in healthy volunteers by NMDA antagonists (e.g., phencyclidine) [136], cognitive-impairing effects of which parallel those seen in schizophrenia clinically [137]. 

Interestingly, NMDA antagonists may decrease the GABA-ergic inhibition and thus lead to the release of glutamate and acetylcholine, which in turn induces schizophrenic symptomology [138]. Moreover, modulating the glutamatergic system by the glycine modulatory site (GMS) of the NMDA receptor may help reduce psychotic and cognitive symptoms of schizophrenia, especially by indirect modulation of GMS. For instance, indirect enhancement of synaptic D-serine via the modulation of D-amino acid oxidase consequently normalizes NMDA receptor hypofunction and reduces cognitive impairments [139]. Likewise, an FDA-approved antipsychotic lumateperone is an antagonist for 5-HT2A receptors that also modulates dopamine and glutamate receptors [140,141].

Another promising target group for the treatment of schizophrenia is a family of trace amine-associated receptors (TAARs). For example, TAAR1 agonists modulate presynaptic pathways and regulate dopamine- and glutamatergic neurotransmission in schizophrenia, also reducing negative symptoms and improving cognitive functions in rodent and primate models of this disorder [142]. Specifically, TAAR1 agonists inhibit the dopaminergic pathways in midbrain, enhance glutamatergic circuits in the prefrontal cortex, and also regulate central serotonergic system [142]. Notably, TAAR1 agonists not only treat positive symptoms of schizophrenia, but also ease its negative symptoms and cognitive impairments [143]. For instance, SEP-363856 (a serotonin 5-HT1A receptor and TAAR1 modulator) shows promising results decreasing schizophrenic symptoms clinically [144].

The interplay between the monoaminergic and the cholinergic systems in schizophrenia is also observed, since schizophrenic patients show a loss of 75% of muscarinic M1 receptors [145]. Drugs binding to M1 receptors improve cognitive functions in rodents, and some of them show promise in clinical practice (e.g., KarXT, acting via muscarinic receptors, reduces cognitive and positive symptoms) [146]. Furthermore, serotonergic 5-HT2A hyperactivity [147] caused by stress, especially in the anterior cingulate cortex and dorsolateral frontal lobe, leads to synaptic atrophy and loss of the gray matter. A novel atypical antipsychotic, pimavanserin, is an agonist at 5-HT2A receptors that reduces psychotic symptoms, especially in Alzheimer patients [148]. Likewise, pharmacogenetic factors also contribute to the pathogenesis and development of personalized medicines for schizophrenia. For example, since inhibitory GABA interneurons contribute to pathogenesis of schizophrenia [149], the glutamate decarboxylase (GAD) and the GABA membrane transporter-1 (GAT) genes are downregulated in schizophrenic patients [150].

As with depression, neuroimmune mechanisms play a key role in pathogenesis of psychoses. For example, microglia promote the degradation of gray matter in schizophrenic patients and reduce neuroprotection by BDNF [151]. In turn, activated microglia (via proinflammatory cytokines) induce neuronal apoptosis [152] and neuroinflammation [153], as, for example, is often seen in postmortem brain samples from schizophrenic patients [154]. Moreover, while LPS induces morphological changes and activates microglia and macrophages in the brain [155], immune-based therapeutics have been tested in clinical trials, targeting p38 MAP kinase (losmapimod) [156], COX2 (celecoxib), adjunctive to reboxetine [157] and TNF (infliximab) [82]. Likewise, while stress activates glucocorticoids and consequently reactivates microglia [158], schizophrenic patients display a hyper-functioning neuroendocrine hypothalamo-pituitary-adrenal (HPA) axis [159,160,161] those deficits may precede the first-episode psychosis [160,162,163]. Furthermore, calprotectin, a neuroinflammatory glial marker, is increased in schizophrenic patients [164]. Finally, some patients with schizophrenia display elevated levels of proline and the proline dehydrogenase (*PRODH*) gene over-expression [165], hence implicating abnormal proline metabolism in schizophrenia. In line with this, administration of proline to zebrafish (*Danio rerio*) triggers schizophrenia-like states in this aquatic model, whereas a neuroleptic sulpiride (but not haloperidol) protects from them [166].

Another interesting candidate novel antipsychotic drug is ulotaront, a mixed TAAR1 and 5-HT1A receptor agonist that is chronically efficient in patients with acute schizophrenia [145]. MK-8189 is a potent and highly selective inhibitor of PDE10A (an important regulator of striatal signaling that, when inhibited, can normalize dysfunctional activity) currently being developed as a novel therapeutic for schizophrenia [167]. Furthermore, cannabidiol (CBD) has been tested as an adjunct treatment to antipsychotics. For example, individuals with schizophrenia receiving CBD (1000 mg) for six weeks have fewer positive psychotic symptoms than placebo [168], thus implying some beneficial effects of CBD in patients with schizophrenia.

## 4. In Silico-Driven Search for Novel Therapeutic Agents

Modern drug development actively employs computer-aided drug design (CADD) methods in the search for novel therapeutic agents and drug targets. CADD-based approaches are traditionally divided into target- and ligand-based drug designs [169]. Target-based drug design (e.g., docking) utilizes 3D structures of drug targets related to the treatment of respective disorders. Ligand-based drug design, based on the knowledge of structures and experimental data on ligands tested in interactions with the drug targets, most commonly includes the similarity estimation and structure–activity relationships ((Q)SAR) models. Since both CADD methods require knowing molecular targets for their respective disorders, the discovery of novel drug targets is a necessary prerequisite for the search for new effective drugs, typically performed using bioinformatics and systems biology (e.g., OMICS) data [169,170].

Over the last decade, there has been a rapid increase in CADD-based studies of depression, including docking studies of ligands for serotonin reuptake [171,172], MAO A and MAO B [173,174], dual action on MAO-B/AChE [175], glycogen synthase kinase [176], sodium hNaV1.2 or hNaV1.7 channels [177], serotonin receptors (5HT1A, 5-HT2A, 5-HT2C and 5-HT4) [171,178,179,180,181], adenosine A1/A2A receptors [182], T-type calcium channels [183], tryptophan 2,3-dioxygenase [184] and sigma receptor [185]. Similarly, application of docking in psychoses involved ligands for serotonin 5HT2 and dopamine D2 receptors [186], α4β2 and α7 nicotinic acetylcholine receptors [187,188], phosphodiesterase 10A [189], MAO A and B [190], a syntaxin-binding protein (STXBP1) [191], NMDA type subunit 1 (GRIN1) [192], fatty acid binding protein 7 (FABP7) [193,194], metabotropic glutamate mGluR5 receptor [195], ionotropic GABA-A receptor [196], glycine transporter type 1 (GlyT1) [197] and kynurenine aminotransferase II (KATII) [198].

CADD strategy may also involve natural compounds and probing pharmacological effects of their extracts, combining a network pharmacology approach and docking. In general, network pharmacology utilizes the systems biology methods to analyze biological networks (e.g., metabolic or signaling pathways, protein–protein interactions) in order to infer drug actions and interactions with various targets [199]. Multiple recent studies have revealed drug targets of phytocomponents from extracts with antidepressant effects [193,194,200,201,202], and a similar approach has been used to search for drug targets related to the treatment of schizophrenia by known schizophrenia drugs [203]. Some studies also combine docking with (Q)SAR methods, e.g., identifying monoamine neurotransmitters reuptake inhibitors as antidepressants [204] or a selective positive allosteric modulation of α1-containing GABA-A receptors [196]. The use of only QSAR models, albeit less common than docking studies, has linked antidepressant effects to MAO A [205], serotonin 5-HT2A receptor [206] and norepinephrine/dopamine reuptake activity [207], and antipsychotic effects - to 5-HT6 [208], D2, 5-HT2A [209] and sigma-2 receptors [210].

There are freely available web services and applications that facilitate the search for possible ligand–target interactions based on the structural formula of compounds. These useful tools are based on similarity estimation (e.g., SwissTargetPrediction [211]), SAR models (e.g., PASS Online [212,213], Super-PRED [214]) and docking (e.g., [215], 1-Click Docking [216]); also see [217] for details. For example, the PASS Online database can predict not only the action on molecular targets, but also the associated pharmacological effects. Briefly, if there is a simultaneous prediction of molecular mechanisms of action of the compound and the corresponding pharmacological effect, the chance to corroborate this effect in the experiment increases significantly, since this confirms the action of the substance at different (molecular, cellular, tissue/organ, and the whole organism) levels of biological organization.

Such knowledge of mechanism–effect relationships, extracted from the literature, is implemented in the PharmaExpert software developed for interpreting the PASS prediction results and containing >15,000 such relationships [169]. The PASS and PASS Online (version 2022) databases predict antidepressant effects with the invariant accuracy of prediction (IAP, equivalent to an area-under-the-curve/AUC value and calculated by the leave-one-out cross-validation (LOO CV) procedure) of 0.897, yielding 90 related mechanisms of actions with the mean accuracy of prediction of ~0.977 (Appendix A). Evaluation by the PASS software of pharmacological potential of phytocomponents from St John’s wort (*Hypericum perforatum*) and chaff-flower (*Achyranthes aspera),* the two well-known medicinal plants with established antidepressant effect, has also been performed [218,219]. Computational analyses of St John’s wort extract activity assessed the predicted biological activity spectra for 93 phytocomponents, revealing several likely phytocomponents that may be responsible for its pharmacological (e.g., antidepressant) effects [210]. Studying eight phytocomponents from chaff-flower predicts their likely antidepressant profile, with estimated probabilities exceeding those of conventional antidepressants. Notably, such simultaneous prediction of both antidepressant effects and the putative mechanisms of action markedly facilitates CNS drug screening, as for the chaff-flower extract that was experimentally tested in animal models and did show antidepressant-like effects [211].

Antipsychotic profile is predicted with the accuracy of 0.910, and 69 related mechanisms of actions are predicted with mean accuracy of prediction 0.983 (Appendix A). Because the accuracy of prediction of pharmacological effects is less than that for molecular mechanisms of action, simultaneous prediction of the pharmacological effect and the associated mechanisms of action is important for experimental validation. The latest version of PASS Online (http://way2drug.com/all/, accessed on 1 April 2023) enables selecting pharmacological effects and appropriate mechanisms of action based on PharmaExpert data, as seen for key activities predicted by the PASS Online database related to antidepressant and antipsychotic activity (Table 1 and Table 2).

Furthermore, predicting biological activity spectra for substances and the knowledge of mechanism–effect relationships for antidepressant and antipsychotic effects provide an opportunity to study not only individual drugs but also drug combinations and complex phytocomponents. This may help reveal the most promising candidates that act via distinct pathogenetic mechanisms, hence leading to synergistic therapeutic effects with, possibly, fewer side effects.

## 5. Conclusions

Complementing traditional targets for pharmacological treatment of depression and schizophrenia (Figure 1), novel putative drug targets continue to emerge, implicating a wide range of CNS mechanisms and molecular circuits (Figure 2). However, in addition to numerous questions that remain open in this field (Table 3), other challenges continue to factor in. For example, comorbidity states and poorly identified, often overlapping clinical and preclinical symptoms markedly complicate the development of new drugs and their practical use for the treatment of depression and schizophrenia. Likewise, as already mentioned, there seem to be several common, overlapping molecular targets for both antidepressants and antipsychotics (Figure 1). As such, it is logical to expect that novel CNS drugs can be developed that target both disorders simultaneously via those common ‘shared’ molecular targets (Table 3). For example, from a conceptual standpoint, it is plausible that novel ‘combined action’ antidepressant or antipsychotic drugs may be developed based on simultaneous targeting of more than one aberrant signaling system (e.g., GABA + serotonin, TAAR + serotonin, dopamine + glutamate). However, if they do this, potential risks and benefits of using such a pharmacotherapeutic strategy are not fully understood, warranting further pre-clinical (and, eventually, clinical) testing.

Because neuroplasticity plays an increasingly recognized role in the pathogenesis of depression and schizophrenia (Figure 1), modulation of CNS remodeling may become a promising target for CNS drug discovery. Likewise, given a key role of neuroinflammation in depression, novel antidepressant drug candidates may emerge that can exert ‘combined’ (e.g., neurotropic + anti-neuroinflammatory, or antidepressant + antipsychotic) activity (Table 3), consistent with the idea of complex, polytarget neuropharmacotherapy for CNS pathogenesis. However, again, if they do exert such effects, their potential risks and benefits for CNS pharmacotherapy warrant further studies.

Moreover, neuroimmune factors may also determine drug resistance, as a separate but related trait, as well. For example, high levels of cytokines TNF-α and IL-6 correlate with resistance to SSRIs [226]. However, neuroinflammation can also be a side effect of CNS drugs, for instance, as some antidepressants may trigger neuroinflammation and activate microglia [227]. Pro-inflammatory cytokines, in turn, may modulate central glutamatergic and monoaminergic systems, neurotrophins, neurohormones, and cellular immune cascades [92,228,229]. Thus, the link between the immune and the nervous systems emerges as an important potential target for novel CNS drugs. Immunological bases of depression are actively investigated both clinically and in animal models [230], showing marked symptomatic similarity between species [231,232,233,234,235,236,237,238]. 

Likewise, the melanocortin system, stress resilience and sleep/wake patterns are also important for normal brain functioning, and their deficits can trigger both depression and psychoses. As oxidative stress triggers neuroinflammation, developing novel antioxidants may also be promising for treating psychiatric disorders. Mitochondrial deficits cause neuronal network damage and, hence, trigger affective and schizophrenic symptoms [239]. Apoptosis is also observed in some brain areas in schizophrenic patients [240], necessitating further studies of antipsychotic (and, possibly, antidepressant) potential of antiapoptotic drugs (Table 3).

Lentivirus and adeno-associated viruses are another strategy for probing antidepressant and antipsychotic mechanisms in the brain via targeted activation or inactivation of gene expression of specific drug receptors, transporters [227,228,229] or nanocarriers (e.g., clozapine targeting 5-HT1A and D2 receptors due to its low selectivity) [241]. Considering individual characteristics of patients is also critically important for the success of psychopharmacotherapy. For example, sex differences (Table 3) are widely discussed in the context of psychiatric diseases [242,243], and some antipsychotics and antidepressants show sex differences in clinical pharmacokinetics [226,244] as well as in animal models [245,246]. 

Further broadening the spectrum of potential molecular drug targets beyond obvious well-established neurochemical systems is becoming critically important as well. For instance, in addition to its well-established physiological role as a regulator of Ca^++^ metabolism and bone growth, vitamin D has emerged as a potent neurosteroid hormone (Table 3) whose deficiency and aberrant signaling via nuclear vitamin D receptors (VDRs) have been linked to both depression and psychoses clinically, as well as in animal models [247,248,249]. This raises the possibility that novel CNS drugs can possibly be developed based on targeting the CNS vitamin D/VDR signaling system.

Finally, further broadening the spectrum of model organisms beyond traditional (e.g., rodent) models is necessary for innovating antidepressant and antipsychotic drug discovery (Table 3). For example, mounting evidence shows that relatively novel ‘alternative’ model organisms like zebrafish can be used in neuroscience not only to generate genetic, pharmacological or other experimental models of human CNS disorders, but to screen for a wide spectrum of CNS drugs as well, including both antidepressants [5,6] and antipsychotics [166]. Characterized by robust face, predictive and construct validity, such models offer genetic tractability, high genetic and physiological homology and an unparalleled high-throughput drug screening capacity [5] that can collectively foster the search and development of novel antidepressant and antipsychotic drugs. Addressing this and other remaining problems and questions (Table 3) can be expected to advance innovative antidepressant and antipsychotic drug discovery and promote further personalizing pharmacotherapy for depression and schizophrenia.

## Figures and Tables

**Figure 1 ijms-24-09482-f001:**
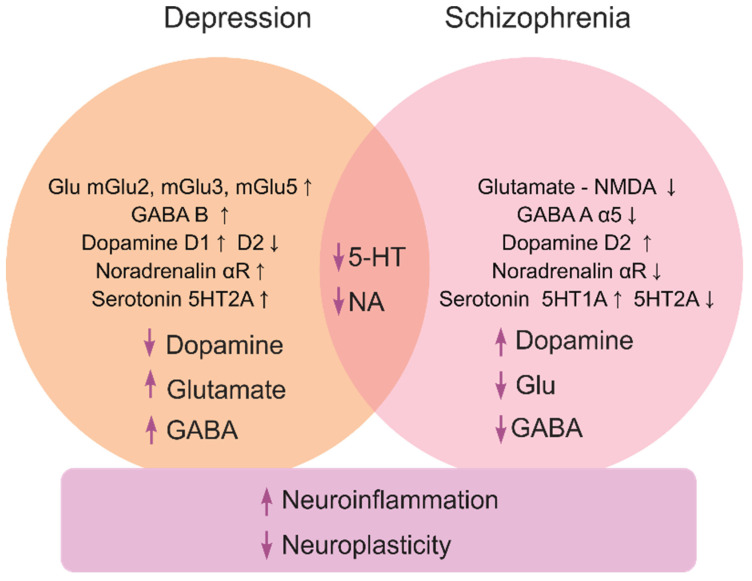
A brief summary of major molecular mechanisms (at both the receptor and neurotransmitter levels) underlying depression and schizophrenia pathogenesis, and their related processes.

**Figure 2 ijms-24-09482-f002:**
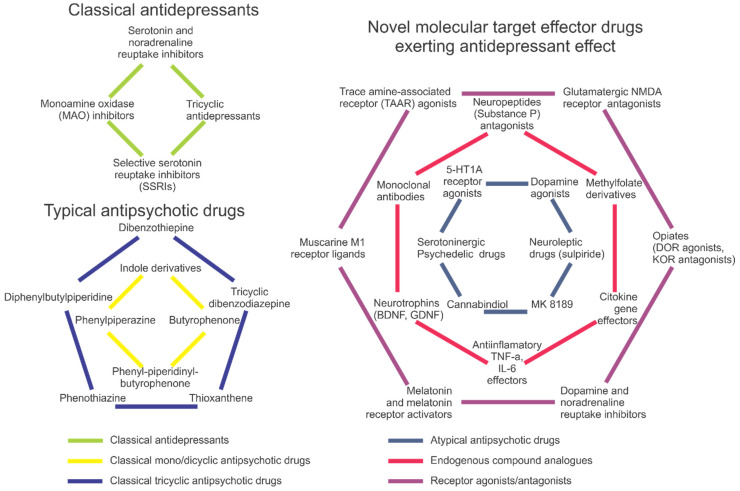
A brief summary of antidepressant and antipsychotic drugs, including conventional (typical) antidepressants and antipsychotics (**left** panel) and various novel atypical and newest drugs with antidepressant and antipsychotic properties (**right** panel).

**Table 1 ijms-24-09482-t001:** Selected active compounds and their invariant accuracy of prediction (IAP) for antidepressant profile and related key mechanisms of action, as predicted by the PASS Online 2022 database.

No	The Number of Active Compounds with the Respective Activity	IAP Based on Leave-One-Out Cross-Validation *	Predicted Activity Profile
1	19,174	0.897	Antidepressant
2	3101	0.989	Serotonin (5 Hydroxytryptamine) 1 agonist
3	1701	0.991	5 Hydroxytryptamine 1A agonist
4	5764	0.984	5 Hydroxytryptamine 1A antagonist
5	135	0.989	5 Hydroxytryptamine 1B agonist
6	7461	0.968	5 Hydroxytryptamine 2 antagonist
7	5262	0.979	5 Hydroxytryptamine 2A antagonist
8	2548	0.988	5 Hydroxytryptamine 6 antagonist
9	1272	0.985	5 Hydroxytryptamine 7 antagonist
10	6367	0.984	5 Hydroxytryptamine agonist
11	18,747	0.967	5 Hydroxytryptamine antagonist
12	7398	0.985	5 Hydroxytryptamine uptake inhibitor
13	244	0.997	AMPA receptor agonist
14	4131	0.983	Adrenaline uptake inhibitor
15	2759	0.973	Alpha 2 adrenoreceptor antagonist
16	2896	0.983	Dopamine agonist
17	3932	0.985	Dopamine uptake inhibitor
18	1112	0.966	GABA receptor agonist
19	1212	0.996	Glutamate (mGluR2) antagonist
20	312	0.993	Glutamate (mGluR3) antagonist
21	2623	0.972	MAO A inhibitor
22	3993	0.977	MAO B inhibitor
23	5366	0.964	MAO inhibitor
24	593	0.994	Melatonin agonist
25	929	0.987	NMDA 2B receptor antagonist
26	27	0.999	NMDA receptor glycine site B antagonist
27	731	0.997	NMDA receptor glycine site antagonist
28	434	0.983	Nicotinic alpha4beta2 receptor antagonist
29	3884	0.970	Opioid kappa receptor antagonist

* equivalent of an area under the curve (AUC) value.

**Table 2 ijms-24-09482-t002:** Selected active compounds and their invariant accuracy of prediction (IAP) for antipsychotic profile and related key mechanisms of action, as predicted by the PASS Online 2022 database.

No	The Number of Active Compounds with the Respective Activity	IAP Based on Leave-One-Out Cross-Validation *	Predicted Activity Profile
1	48	0.910	Antischizophrenic
2	7461	0.968	Serotonin (5 Hydroxytryptamine) 2 antagonist
3	5262	0.979	5 Hydroxytryptamine 2A antagonist
4	2432	0.986	5 Hydroxytryptamine 3 antagonist
5	2548	0.988	5 Hydroxytryptamine 6 antagonist
6	1272	0.985	5 Hydroxytryptamine 7 antagonist
7	1835	0.992	Acetylcholine M1 receptor agonist
8	411	0.997	Acetylcholine M4 receptor agonist
9	1604	0.979	Acetylcholine nicotinic agonist
10	591	0.997	Dopamine D1 agonist
11	8375	0.983	Dopamine D2 antagonist
12	3789	0.984	Dopamine D3 antagonist
13	2387	0.986	Dopamine D4 antagonist
14	10,756	0.980	Dopamine antagonist
15	588	0.996	Estrogen receptor beta agonist
16	885	0.994	Glutamate (mGluR2) agonist
17	114	0.993	Glutamate (mGluR3) agonist
18	1999	0.996	Glycine transporter 1 inhibitor
19	6006	0.977	Glutamate NMDA receptor antagonist
20	975	0.988	Nicotinic alpha7 receptor agonist
21	5287	0.992	Phosphodiesterase 10A inhibitor
22	629	0.970	Trace amine-associated receptor 1 agonist

* equivalent of an area under the curve (AUC) value.

**Table 3 ijms-24-09482-t003:** Selected potential open questions related to developing novel antidepressant and antipsychotic therapies.

** *General conceptual questions:* ** Are there common, overlapping molecular targets for both antidepressants and antipsychotics? (also see Figure 1). Can novel central nervous system (CNS) drugs be developed that target both disorders simultaneously? Can novel antidepressant or antipsychotic drugs be developed based on simultaneous targeting of more than one aberrant signaling system (e.g., GABA + serotonin, dopamine + glutamate)? What are potential risks and benefits from such approaches? Environmental factors play a role in shaping clinical, including genetically determined, depression and schizophrenia (i.e., the gene x environment interactions). Similarly, environmental enrichment alleviates depression- and schizophrenia-like behaviors in animal models [220,221,222]. Can such environmental factors influence the efficacy of novel antidepressant and antipsychotic drugs, and how can this be assessed in pre-clinical and clinical studies? Epigenetic factors play an important role in CNS pathogenesis, including both affective and psychotic illnesses. Can novel CNS drugs be developed (and, eventually, introduced and approved) based on targeting epigenetic mechanisms in the brain? What is the exact role of neuronal vs. neuroglial (and also microglial vs. astrocytic) mechanisms in depression and schizophrenia? Can novel drugs be developed based on specific targeting of such cell type-specific processes? Are there common and *disorder-specific* contributions from glial cells in depression and schizophrenia? Can novel CNS drugs be developed based on targeting those putative common (shared) and disorder-specific processes? Are there common and *disorder-specific* neurogenomic, neuroproteomic and neurometabolomic signatures of depression and schizophrenia? Can novel drugs be developed based on this omics information? ** *Selected specific biomedical questions:* ** What is the exact role of brain-derived neurotrophic factor (BDNF) and its signaling pathways in modulating depression and schizophrenia? Can novel antidepressant and antipsychotic drugs, and potentially ‘combined action’ CNS drugs, be developed based on targeting BDNF and other brain neurotrophins? What is the role of neuronal and neuroglial apoptosis in depression and schizophrenia? Can novel CNS drugs be developed for these two disorders based on targeting apoptosis? Inflammatory cytokines can induce aberrant mTOR activity (e.g., interleukins (IL) IL-1β, IL-17A and TNF-α strongly activate the mTOR kinase PRAS40 and the downstream targets of mTOR activity, 4E-BP1 and the ribosomal protein S6). Can drugs that modulate such cytokines be used for the treatment of depression and psychoses (e.g., by normalizing glutamate signaling indirectly, via the mTOR-dependent processes)? Central trace amines and their receptors (TAARs) have been linked to both depression and psychoses clinically, as well as in animal models. Can novel CNS drugs be developed based on targeting various TAARs? Sex differences have been reported for pharmacological treatment of both depression and schizophrenia [223,224]. How can novel drugs address this clinical aspect, to more precisely target these respective CNS disorders in clinical and pre-clinical studies? For example, can novel steroid-based drugs be novel putative antidepressant and antipsychotics? Certain pro-psychotic drugs (e.g., a deliriant hallucinogenic agent scopolamine) may evoke antidepressant effects [225]. Is there a potential therapeutic value of cross-disorder overlap between drugs modulating depression and schizophrenia? Vitamin D is a potent neurosteroid hormone whose deficiency has been linked to both depression and psychoses clinically, as well as in various animal models. Can novel CNS drugs be developed based on targeting the vitamin D signaling system in the brain? What is the role of nuclear vitamin D receptors (VDRs) in clinical and preclinical depression and schizophrenia? Can novel CNS drugs be developed based on targeting the VDRs? ** *Selected translational questions:* ** Are there potential reliable peripheral biochemical biomarkers of depression and schizophrenia in animal models and/or clinical studies that can be used for fostering CNS drug discovery? How much do non-pharmacological interventions (e.g., diet, physical exercise, cognitive behavioral therapy) contribute to better performance of pharmacological therapy in patients with depressive and schizophrenic patients? How can animal models contribute to our better understanding of this particular aspect? Real-world evidence and real-world data have been used to support clinical trial designs and observational studies to generate new treatment approaches. How do such data facilitate the development of novel antidepressant and antipsychotic drugs? How can animal models contribute to increasing the reliability of these data? ** *Methodological, technical and practical questions:* ** There are well-reported overt strain differences in animal (e.g., mouse) models of depression and schizophrenia. Can meaningful biological information be gathered from such differences that may inform CNS drug search (e.g., can genomic or neurochemical strain differences be translated in pathway differences for these two disorders)? Can novel antidepressant and antipsychotic drugs be developed based on drug repurposing? How does gut microbiota influence depression or schizophrenia states? Can novel probiotics be developed (and, eventually, introduced and approved) as potential antidepressants and antipsychotics? How do various dietary factors (e.g., Mediterranean vs. Western diets) influence depression and schizophrenia? Can novel food supplements be developed (and, eventually, introduced and approved) as potential antidepressants and antipsychotics? How does diet (e.g., high-carbohydrate diet) that predisposes to inflammation impact depression and schizophrenia? Can novel CNS drugs be developed based on reducing diet-promoted inflammation? How can novel computer technologies (e.g., artificial intelligence) accelerate the development of novel antidepressants and antipsychotics? How can data obtained from animal models contribute to increasing the reliability and applicability of these technologies in CNS drug discovery? How can novel alternative model organisms (e.g., zebrafish, Danio rerio) be used to promote innovative CNS drug screening for novel antidepressant and antipsychotic drugs? ** *Other related open questions:* ** Regulatory agencies have sought to reduce the use of animals in the development of novel drugs (e.g., US FDA has recently approved the non-mandatory use of animals before human drug trials). How can these trends and policy shifts impact the development of new antidepressants and antipsychotics immediately, and in the long run? What are specific molecular mechanisms for CNS drug resistance? Can novel antidepressant or antipsychotic drugs be developed based on specific targeting of such ‘drug resistance’ targets? In other words, can such putative new group of drugs be used to specifically prevent or manage treatment-resistant forms of CNS disorders?

## Data Availability

Not applicable.

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
