# Peer review of "Towards Novel Potential Molecular Targets for Antidepressant and Antipsychotic Pharmacotherapies"

_ijms, 2023, doi:10.3390/ijms24119482_

Round 1

Reviewer 1 Report

The authors have reviewed the major molecular targets used in the treatment of depression and schizophrenia. The review is comprehensive enough to be useful to a wide range of readers.

As there are now so many reviews, it is important to indicate in the abstract and introduction what the main novelty of the approach is. For example, it remains unclear how great the need is for drugs that are both antidepressants and antipsychotics. What do the authors see as the advantage of such drugs? The discussion of this question needs to be expanded, and then the relevance and novelty of this review will be clear.

Other comments: 

1. Figure 1: The authors indicate changes in neuromediators, but this is an oversimplified representation that may not be correct in some cases. For example, in schizophrenia, the authors indicate that GABA decreases (second column, top) and GABA increases (third column, bottom). It is better to report changes in the expression of specific molecular targets (receptors) involved in depression and schizophrenia.

2 Authors often use the term "neuroplasticity," but it is not always clear what is meant. It may be synaptic plasticity, membrane plasticity, or other forms. In any case, the characteristics of each form of neuroplasticity and its mechanisms are different, so a more precise description is needed.

3. A number of sections of the review are written very briefly and are therefore of little use. Such sections should either be expanded or deleted. For example, 

p. 7: "GABA is a key inhibitory neurotransmitter [63], exerting its effects via the GABA-A, -B and -C receptors [64]. Interestingly, the overexpression of GABA-related genes is observed in depressed patients [65], while stress reduces BDNF levels, leading to the downregulation of the GABA-ergic system and to mood deficits [66], thus implicating GABA in both neuroplasticity and potential complex multimodal antidepressant therapy."

It is unclear which genes are overexpressed and which are downregulated. GABAc receptors are currently classified as GABAa-rho. Since GABAa and GABAb receptors belong to different classes of receptors (ionotropic and metabotropic), this may also be worth mentioning.

4. There are unfortunate expressions: "Mounting evidence suggests neuroinflammation as a biomarker..." A biomarker is a kind of indicator, a molecule, not the process itself.

"Another treatment strategy is selective delivery of drugs, including using lentivirus and adeno-associated viruses [230-232]." Viral constructs deliver genes but not drugs. It is worth writing more clearly what is meant.

5. Table 3 contains a considerable number of questions. However, the logic of their appearance in the table is unclear. For example, there is a question about vitamin D that is not discussed in the review. It is recommended that only key questions remain in the table.

Minor editing of English language required

Author Response

Editorial Office

IJMS

May 13, 2023

Re: MS by Kositsyn et al.

Dear Editorial Office,

Thank you very much for your e-mail regarding our manuscript by Kositsyn et al., submitted to your Journal. We thank the Editor and expert Reviewers for their helpful comments and a generally positive evaluation of our paper.

As requested, please find attached our revised manuscript and the point-by-point rebuttal letter that addresses the Reviewer’ concerns. For your convenience, all revised items are now highlighted in yellow in the resubmitted MS text file.

 
Reviewer comments:

Reviewer #1

Point 1: As there are now so many reviews, it is important to indicate in the abstract and introduction what the main novelty of the approach is. For example, it remains unclear how great the need is for drugs that are both antidepressants and antipsychotics. What do the authors see as the advantage of such drugs? The discussion of this question needs to be expanded, and then the relevance and novelty of this review will be clear.

Reply: The authors thank Reviewer 1 for this point. To address the expert Reviewer’s suggestions, we have modified the Introductory sections of our paper, to better emphasize the novelty of our MS’ foci and its outlining of multiple strategic developments in the field (also see Table 3), as well expanded upon several topics, as recommended by Reviewer 1.

Point 2. Figure 1: The authors indicate changes in neuromediators, but this is an oversimplified representation that may not be correct in some cases. For example, in schizophrenia, the authors indicate that GABA decreases (second column, top) and GABA increases (third column, bottom). It is better to report changes in the expression of specific molecular targets (receptors) involved in depression and schizophrenia.

Reply: We agree with this comment, as the point is indeed well-taken.  To address the expert Reviewer’s critique, we added the main neuromediator receptor changes (upregulation or downregulation) in schizophrenia and depression. While the levels of some neuromediators in schizophrenia and depression undergo time-associated changes, Figure 1 also shows the predominant major level of the respective mediators.  The figure legend has been modified to this effect as well, aiming at a greater clarity.

Point 3. Authors often use the term "neuroplasticity," but it is not always clear what is meant. It may be synaptic plasticity, membrane plasticity, or other forms. In any case, the characteristics of each form of neuroplasticity and its mechanisms are different, so a more precise description is needed.

Reply: We fully agree with this comment, and have now clarified the intended neuroplasticity form in the text, as requested.

Point 4. A number of sections of the review are written very briefly and are therefore of little use. Such sections should either be expanded or deleted. For example, on GABA. It is unclear which genes are overexpressed and which are downregulated. GABAc receptors are currently classified as GABAa-rho. Since GABAa and GABAb receptors belong to different classes of receptors (ionotropic and metabotropic), this may also be worth mentioning.

Reply: We have modified and expanded this section, also citing additional references, to address the Reviewer’s concerns.

Point 5.  There are unfortunate expressions: "Mounting evidence suggests neuroinflammation as a biomarker..." A biomarker is a kind of indicator, a molecule, not the process itself.

Reply: Corrected, as requested.

Point 6. Likewise, "Another treatment strategy is selective delivery of drugs, including using lentivirus and adeno-associated viruses [230-232]." Viral constructs deliver genes but not drugs. It is worth writing more clearly what is meant.

Reply: Corrected, as requested, to state that lentivirus and adeno-associated viruses are another treatment strategy to enhance the effect of antidepressants and antipsychotics activating gene expression of specific drug receptors, aiming at a greater clarity.

Point 7. Table 3 contains a considerable number of questions. However, the logic of their appearance in the table is unclear. For example, there is a question about vitamin D that is not discussed in the review. It is recommended that only key questions remain in the table.

Reply: We thank Reviewer 1 for their helpful constructive comments. To attend to these peer review concerns, we have now modified and restructured Table 3, aiming at a greater organization (from larger/broader to more specific) of open questions, in order to improve their logical flow. We also slightly expanded upon the role of Vitamin D role in the MS text – but could only do so briefly, since the aim of Table 3 was no highlight some future potential topics and lines of research, sometimes including those topics for which there is no much literature currently available.

Overall, the authors believe that in the revised MS we have now carefully addressed all concerns and comments made by expert Reviewer. We shall be grateful if you could consider our revised MS for publication in your Journal.

Sincerely,

Authors

Reviewer 2 Report

This is an interesting review on  novel potential molecular targets for antidepressant and antipsychotic pharmacotherapies The review covers a number of potential drug targets however some modifications should be introduced before publication

 Abstract – is too general, the major directions should be mentioned

 Figure 1 A similar review by WieroÅ„ska, et al., Depression and schizophrenia viewed from the perspective of amino acidergic neurotransmission: Antipodes of psychiatric disorders Pharmacology and Therapeutics, 193, pp. 75-82.) was published

 Figure 2  letters are too small and hard to read

 The subject of mGlu receptors in depression/schizophrenia should be covered in a more detailed manner

 The first papers showing the antidepressant efficacy of scopolamine should be mentioned (Furey, M.L., Drevets, W.C. (2006) Archives of General Psychiatry, 63 (10), pp. 1121-1129.

 Minor editing of English language required

Author Response

Editorial Office

IJMS

May 13, 2023

Re: MS by Kositsyn et al.

Dear Editorial Office,

Thank you very much for your e-mail regarding our manuscript by Kositsyn et al., submitted to your Journal. We thank the Editor and expert Reviewers for their helpful comments and a generally positive evaluation of our paper.

As requested, please find attached our revised manuscript and the point-by-point rebuttal letter that addresses the Reviewer’ concerns. For your convenience, all revised items are now highlighted in yellow in the resubmitted MS text file.

Reviewer comments:

Reviewer 2

Point 1. This is an interesting review on novel potential molecular targets for antidepressant and antipsychotic pharmacotherapies. The review covers a number of potential drug targets however some modifications should be introduced before publication. Abstract – is too general, the major directions should be mentioned

Reply: Corrected and streamlined, as requested.

Point 2. Figure 1 A similar review by Wierońska, et al., Depression and schizophrenia viewed from the perspective of amino acidergic neurotransmission: Antipodes of psychiatric disorders Pharmacology and Therapeutics, 193, pp. 75-82.) was published

Point 3. Figure 2 letters are too small and hard to read

Reply: Modified, as requested.

Point 4. The subject of mGlu receptors in depression/schizophrenia should be covered in a more detailed manner.

Reply: During the revision, we have now provided a more detailed description of the glutamatergic system in schizophrenia/depression, as per expert Reviewer’s comments. Additional relevant papers have also been cited in the revised MS as well.

Point 5. The first papers showing the antidepressant efficacy of scopolamine should be mentioned (Furey, M.L., Drevets, W.C. (2006) Archives of General Psychiatry, 63 (10), pp. 1121-1129.

Reply: Provided in the revised MS, as requested.

Overall, the authors believe that in the revised MS we have now carefully addressed all concerns and comments made by expert Reviewers. We shall be grateful if you could consider our revised MS for publication in your Journal.

Sincerely,

Authors

Round 2

Reviewer 1 Report

The authors have answered all my questions. There are no further comments.